# Influence of Additive N_2_ on O_2_ Plasma Ashing Process in Inductively Coupled Plasma

**DOI:** 10.3390/nano12213798

**Published:** 2022-10-27

**Authors:** Ye-Bin You, Young-Seok Lee, Si-Jun Kim, Chul-Hee Cho, In-Ho Seong, Won-Nyoung Jeong, Min-Su Choi, Shin-Jae You

**Affiliations:** 1Department of Physics, Chungnam National University, Daejeon 34134, Korea; 2Institute of Quantum Systems (IQS), Chungnam National University, Daejeon 34134, Korea

**Keywords:** plasma ashing, oxygen radical density, optical emission spectroscopy, Langmuir probe, EEPF

## Abstract

One of the cleaning processes in semiconductor fabrication is the ashing process using oxygen plasma, which has been normally used N_2_ gas as additive gas to increase the ashing rate, and it is known that the ashing rate is strongly related to the concentration of oxygen radicals measured OES. However, by performing a comprehensive experiment of the O_2_ plasma ashing process in various N_2_/O_2_ mixing ratios and RF powers, our investigation revealed that the tendency of the density measured using only OES did not exactly match the ashing rate. This problematic issue can be solved by considering the plasma parameter, such as electron density. This study can suggest a method inferring the exact maximum condition of the ashing rate based on the plasma diagnostics such as OES, Langmuir probe, and cutoff probe, which might be useful for the next-generation plasma process.

## 1. Introduction

For decades, reducing the size of the devices has been an essential issue in terms of device functionality and cost in the semiconductor manufacturing industry. Recently, the size of semiconductor devices has been reduced to nano-scale, and accordingly, the semiconductor process has become more detailed. A photoresist is widely used in nano-scale processes because it can be simply used as a mask using light and is also easy to remove [1,2,3,4]. However, photoresist (PR) is only used as a mask for patterning, so it should be removed after use. Typical methods for removing photoresists are wet and dry [5,6,7,8,9]. Wet etching is a method using a chemical solution, and since photoresist is made of polymer, it is removed using an acidic solution. Wet etching is advantageous in cost because it is simple to set up and has a relatively short time compared to dry etching, but it is limited in that it is difficult to analyze sample corrosion and trace elements due to strong acidity [10,11,12]. Therefore, in the nano-scale process, dry etching using plasma is used [13,14,15,16].

The semiconductor process, as if by Moore’s Law, has become refiner and more integrated over a few decades. Accordingly, the number of semiconductor processes has increased significantly, and it has become very important to reduce the processing time. It is necessary to increase the etching rate because dry etching, called plasma ashing, takes a longer process time than wet etching. Since the 1990s, there have been many studies to increase the ashing rate. For example, there are methods of controlling the substrate temperature, changing the gas flow rate, or inserting additional gas into oxygen plasma [17,18,19]. Among them, FC gas, nitrogen, hydrogen, etc., are used to insert additional gas into the oxygen plasma. Additional gases play a role in increasing the concentration of oxygen radical species in oxygen plasma, and nitrogen gas was used in this experiment [20,21]. In the previous study, it was reported that when nitrogen gas is added to oxygen plasma, nitrogen gas combines with oxygen radical species to form NO [22], thereby preventing the oxygen radical species recombination and increasing concentration. In addition, nitrogen plasma has relatively more vibrational and metastable states of nitrogen molecules than oxygen plasma, resulting in more dissociation of oxygen molecules [23]. Nitrogen, as well as other additional gases, similarly exhibit the effect of increasing the concentration of oxygen radical species by dissociating more oxygen molecules. In the ashing process, eventually, the most important factor is the density of oxygen radical species, so it is essential to set the ratio of additional gases to the maximum density of radical species. Then, the optimum point of oxygen radical species density can be found through plasma diagnostics.

There are many methods to diagnose plasma parameters; electron density, electron temperature, plasma potential, radical density, etc. One of them is the Langmuir probe, which measures the current entering the metal tip driving tip voltage by penetrating the plasma [24,25,26]. We can obtain the electron energy probability function (EEPF) with the current by the Langmuir probe and determine electron density, electron temperature, and plasma potential through the EEPF. Moreover, among the methods for measuring the number of radicals in plasma, there is optical emission spectroscopy (OES) using light emitted from the excitation-relaxation process of gas. OES is a method of qualitatively determining the number of gas species by analyzing the spectrum of light emitted from the process. However, the amount of light obtained through OES depends on several variables such as electron density, rate coefficient, etc. Therefore, accurate radical density cannot be obtained by light intensity of OES only. In previous studies, many groups studied methods for converting the intensity obtained through OES into quantitative radical density [27,28,29,30]. One of the most commonly used methods is actinometry. Actinometry is a method of injecting and comparing an additional gas other than process gases, where the ratio of the rate coefficient between the gas and processing gas used should be constant depending on the electron temperature. Although the radical density can be inferred quite accurately using actinometry, there is a disadvantage in that a gas other than the process gas used should be injected [31].

In this study, compared with the previous study, we performed a more comprehensive experiment of the O_2_ plasma ashing process in various N_2_/O_2_ mixing ratios and RF powers; our investigation revealed that the tendency of the density measured using only OES did not exactly match the ashing rate [32]. This problematic issue can be solved by considering the plasma parameter, such as electron density. The detailed experiment procedure, result, and discussion are described in this paper [33,34].

## 2. Experiment

Figure 1 shows the schematic of our experimental setup. The chamber used in the experiment was an inductively coupled plasma source (ICPs), and the experiment was conducted while increasing the RF power of 13.56 MHz to 200–500 W. The ashing process was performed at the chamber pressure of 30 mTorr, and N_2_ gas was injected into O_2_ plasma while increasing to 0–80%. The total flow rate of O_2_ and N_2_ was 150 sccm, and the range of MFC was 500 sccm and 200 sccm, respectively. During the experiment, the substrate of the chamber was cooled to about 20 °C. As shown in Figure 1, a sample of about 1.5 × 1.5 cm^2^ was used in the experiment, and the process was conducted with half the sample covered with Kapton tape to confirm the ashing rate.

In this equipment, ashing starts by turning O_2_ gas injected into the chamber with a flow rate, $R_{O_{2}}$, which leads to the chamber pressure at a certain level, $p_{O_{2}}$, into plasma with an inductively coupled RF power, $P_{RF}$, filling the chamber with O radicals that react with PR to form volatile by-products such as CO and CO_2_. This process can be expressed as,
(1)$$A\propto R_{O_{2}}P_{RF}p_{O_{2}}^{-1}$$
where *A* is the ashing rate, $R_{O_{2}}$ the gas O_2_ flow rate, $P_{RF}$ is the applied *RF* power, and $p_{O_{2}}$ is the O_2_ pressure in the chamber.

Figure 2 is an example of removing the PR deposited on the sample through the ashing process. The PR used in the experiment is AZ-5214 E, positive PR. The photoresist was deposited at 6000 rpm for about 30 s using the spin coating method, and as a result, the deposited thickness was about 1.7 μm. Before and after coating, the sample was baked on a substrate at about 115 °C, pre-baking was used to remove moisture, and later baking was used to solidify PR by removing solvent. The equipment used to measure the ashing rate is Alpha-step D-500, which measures the thickness.

The optical fiber where line-of-sight integrated plasma emissions are detected is laterally mounted on the chamber wall and points to the plasma bulk for strong signal measurement. The wavelength of oxygen radical measured using OES is 8446 Å, also known as the oxygen atom peak [35,36]. Additionally, the tip length of the Langmuir probe used for EEPF measurement is about 3 mm, and the diameter is 0.15 mm. The EEPF was obtained by connecting the Langmuir probe to WiseSLP, which analyzes the I-V curve [37]. The tip length and the diameter of the Langmuir probe used for EEPF measurement are about 3 mm and 0.15 mm, respectively.

## 3. Results and Discussions

Figure 3 shows the ashing rate for each power condition as the ratio of N_2_ to O_2_. The ashing rate has a maximum value at a specific ratio for each condition, with 200 W at 5%, 300 and 400 W at 10%, and 500 W at about 20%, respectively. Additionally, the peak point tended to move toward the higher ratio as the power increased. When nitrogen gas is injected, there are two main reasons for increasing the ashing rate; (i) preventing the recombination of the oxygen radical, (ii) increasing the dissociation of the oxygen molecule [22,23]. In the prevention process, the recombination reaction of oxygen radicals is reduced through the reaction of N + O_2_ → NO + O. Then, the nitrogen atom holds the oxygen radical in the NO form as if it were a carrier. Therefore, it increases the concentration of oxygen radicals by increasing the residual time of oxygen in the radical state. In the process of increasing dissociation, nitrogen in the metastable state and the vibrational state plays that role. In general, nitrogen plasma has more atoms and molecules of metastable and vibrational states, which have higher energy than oxygen plasma. Therefore, in nitrogen plasma, there are more paths for dissociating oxygen molecules than in oxygen plasma. However, if nitrogen gas is injected more, the ashing rate decreases because fewer oxygen molecules can provide oxygen radicals.

In order to compare the tendency of the ashing rate with the density of oxygen radical, the intensity was measured using OES. Figure 4 shows the intensity of oxygen radical with a wavelength of 8446 Å as the power. However, the oxygen radical intensity measured using OES does not follow the tendency of the ashing rate and has a maximum value of 5% under all conditions. The reason why the two tendencies do not match is that the intensity measured using OES has several variables, as well as density [35,38,39]. The intensity depends on the properties of electrons that can excite oxygen radicals, which can be described as follows:(2)$$I_{O}\propto n_{O}n_{e}K\left(T_{e}\right)$$
at $I_{O}\;and\;n_{O}$ are intensity and density of oxygen radicals, respectively; $n_{e}$ is the electron density; K is the rate coefficient of the excitation process; and $T_{e}$ is the electron temperature. Therefore, it is able to know the tendency of the specific oxygen radical density by measuring these variables.

Figure 5 shows the EEPF measured using the Langmuir probe. As shown in the graphs, the EEPF decreased at about 3 eV as the N_2_ gas ratio increased. This is because the collision cross-section of the vibrational state is the highest at 3 eV, which reduces the population of electrons with similar energy, resulting in a decrease in the EEPF [40,41]. The reciprocal of the slope of the EEPF means the electron temperature [42]. Therefore, it may be seen that the temperature of the low-energy electron group of about 3 eV decreases. However, since the excitation energy of oxygen radical is about 10 eV, it corresponds to the high-energy electron group [36,43]. As shown in the graphs, it can be seen that the slope of the high-energy electron group above about 7 eV hardly changes with the N_2_ gas ratio, and the electron temperature was also constant at about 2.4 eV. Therefore, in this study, the electron density was obtained by linear fitting of EEPF of about 7 eV or more, and the equation is as follows,
(3)$$n_{e}=\int_{0}^{\infty }\sqrt{\varepsilon }g_{p}\left(\varepsilon \right)\;d\varepsilon$$
at ε is the electron energy, and $g_{p}$ is the EEPF [44]. Figure 6 shows the electron density. The electron density tended to decrease slightly as the N_2_ gas ratio increased under all conditions.

Then, the tendency of oxygen radical density was inferred by substituting the obtained electron density and electron temperature in Equation (2). It is worth noting that the PR used in this work contains C atoms in its components, and thus oxygen radicals generated in plasmas are expected to play an essential role in removing the PR coating layer from the wafer surface. Figure 7 shows the oxygen radical density and the ashing rate thus obtained. Therefore, when analyzing the tendency of the ashing rate using OES, it is important to measure the electron density. In addition, when modeling intensity to infer radical density, it can be seen that electron density is more important than electron temperature. However, in the result of 500 W in Figure 7, it can be seen that the peak does not exactly match, which may be that UV-influenced PR ashing. In fact, in the OES measurement results, it was confirmed that the UV region intensity of 200–300 nm was increased as N_2_ gas was injected. Therefore, future work is needed on this part.

## 4. Conclusions

In this study, we performed the ashing process using N_2_/O_2_ inductively coupled plasma and measured the light intensity of oxygen radicals by OES. Moreover, we compared the tendency between them depending on the ratio of nitrogen in RF power variation. However, their tendency did not exactly match, and it was assumed that the light intensity was related to plasma parameters such as electron density and temperature. Therefore, we measured the EEPF using the Langmuir probe and obtained plasma parameters from the EEPF. Then, we modeled the relationship between the light intensity and radical density with the plasma parameters. Consequently, we matched the tendency between them more accurately and found that it is necessary to measure the density of the high-energy electron group to infer the oxygen radical density corresponding to the ashing rate trends.

## Figures and Tables

**Figure 1 nanomaterials-12-03798-f001:**
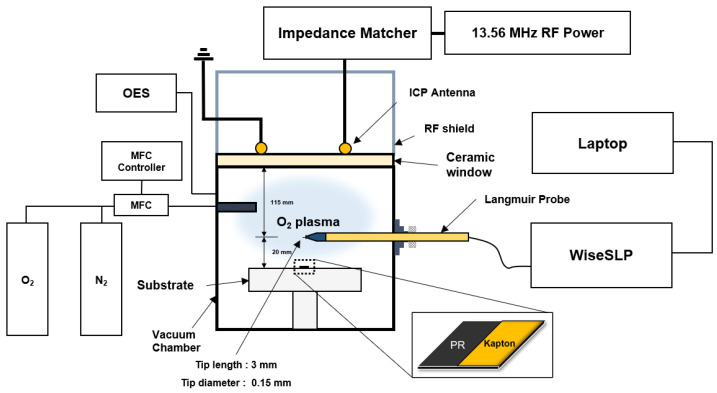
This is a schematic of experimental set-up, the ICPs reactor, and sample.

**Figure 2 nanomaterials-12-03798-f002:**
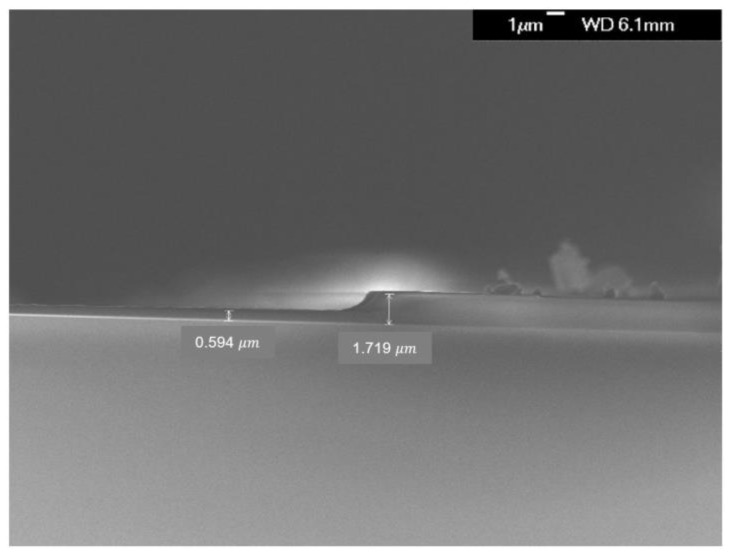
This is an example of removing the PR deposited on the sample.

**Figure 3 nanomaterials-12-03798-f003:**
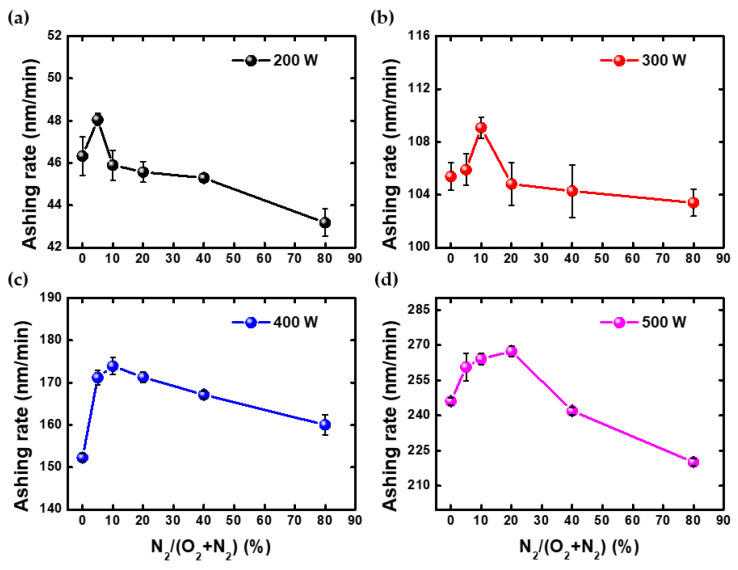
This is the ashing rate as N_2_/(O_2_ + N_2_) ratio at (**a**) 200 W, (**b**) 300 W, (**c**) 400 W, and (**d**) 500 W.

**Figure 4 nanomaterials-12-03798-f004:**
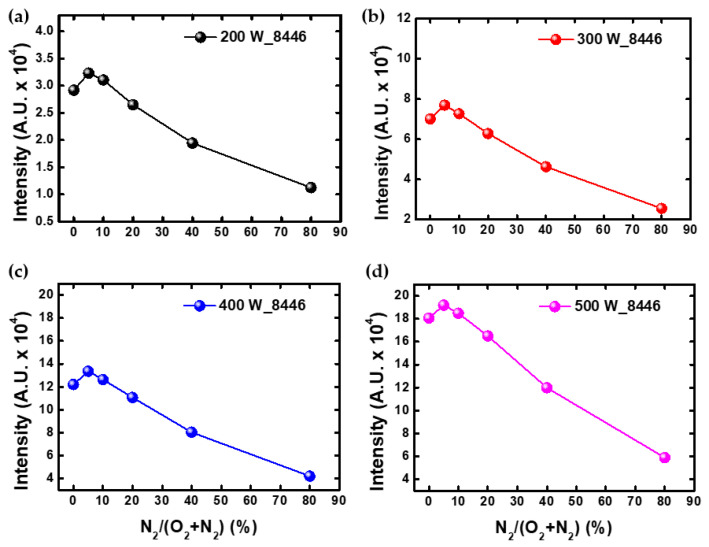
This is the oxygen radical intensity of 8446 Å measured using OES at (**a**) 200 W, (**b**) 300 W, (**c**) 400 W, and (**d**) 500 W.

**Figure 5 nanomaterials-12-03798-f005:**
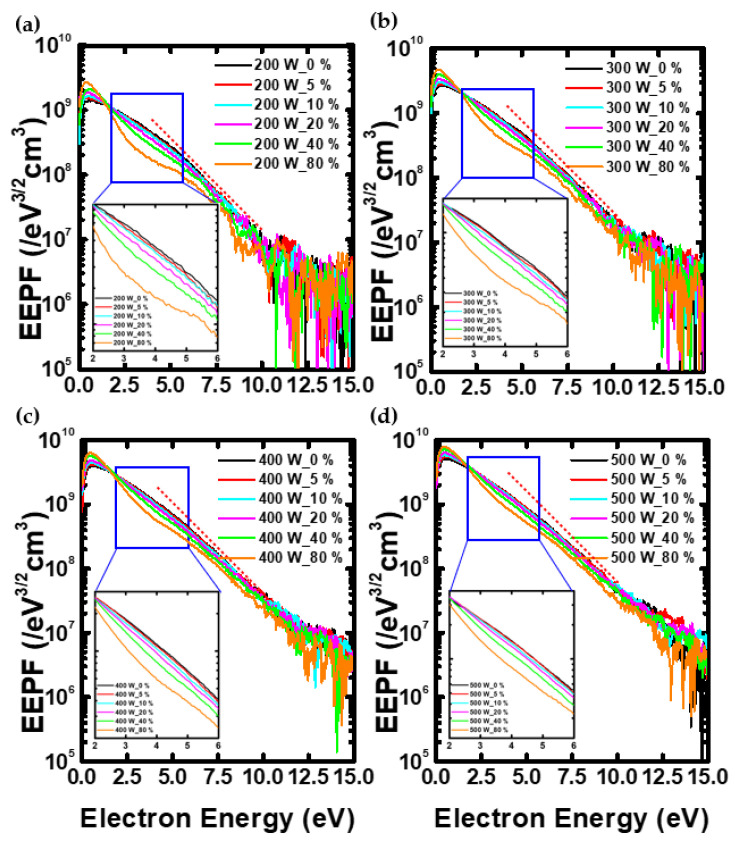
This is the EEPF measured using the Langmuir probe at (**a**) 200 W, (**b**) 300 W, (**c**) 400 W, and (**d**) 500 W.

**Figure 6 nanomaterials-12-03798-f006:**
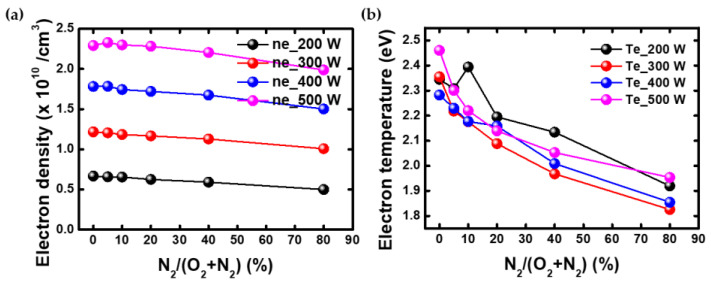
These are plasma parameters obtained by the EEPF as power variation. (**a**) electron density, (**b**) electron effective temperature.

**Figure 7 nanomaterials-12-03798-f007:**
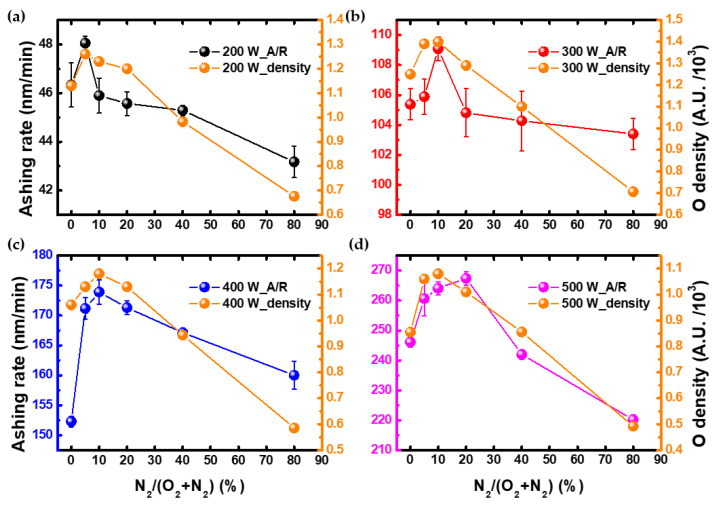
These are the ashing rate and oxygen radical density as N_2_/(O_2_+N_2_) ratio at (**a**) 200 W, (**b**) 300 W, (**c**) 400 W, and (**d**) 500 W.

## Data Availability

The data presented in this study are available on request from the corresponding author.

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
