# Peer review of "Influence of Additive N2 on O2 Plasma Ashing Process in Inductively Coupled Plasma"

_nanomaterials, 2022, doi:10.3390/nano12213798_

Round 1

Reviewer 1 Report

Dear Authors,

I think the paper can be published after the following improvements:

1. The authors should specify the exact expression for the theoretical values of ashing rate , similar to the equation (1). It should be given close to the Figure 1. Explain the general behaviour of the Figure 1 using this expression otherwise it is difficult to follow the authors.

2. Errror. N + O → NO + O in the text...  - >  N + O_2

3. What is the value of the electron temperature measured using EEPF  ? There is no information about it. Present the same figure as in case of the electron density. How does it impact the observed behaviour of the ashing rate.

4. The OES measurements are line-of-sight integrated. What happens if the OES measurements would be performed not at the normal to wafer but along it surfaces ?

5. The ashing rate is increasing as a function of power. But at certain point the high voltage should have also a negative effect Explain it.

6. The results were obtained using the data at 8446 A.. What happens if another spectral line is used  ? Have the authors performed it or not ?

7. Whereas the topic is silghtly different the similar approach is also used to combine the spectroscopic stuies with ashing/sputtering rate in other RF plasmas... see for instance

Sackers et al, Physics of Plasmas 29, 043511 (2022); https://doi.org/10.1063/5.0083613

It would be important to demonstrate the connection between the different topics of plasma and surface physics..

So, If the authors improve and answer the points the paper can be published.

Author Response

Point 1: The authors should specify the exact expression for the theoretical values of ashing rate, similar to the equation (1). It should be given close to the Figure 1. Explain the general behaviour of the Figure 1 using this expression otherwise it is difficult to follow the authors.

Response 1: Thank you for the helpful comment. We fully agree that further description on the ashing equipment would help readers to understand more clearly. Following the comment, we have revised the description about Fig. 1 as follows (as shown in blue in the revised manuscript).

“In this equipment, ashing starts by turning O2 gas injected into the chamber with a flow rate, , that leads to the chamber pressure at a certain level, , into plasma with an inductively coupled RF power, , filling the chamber with O radicals that react with PR to form volatile by-products such as CO and CO2. This process can be expressed as,  

                                                                                                                                   (1)

where A is the ashing rate,  the gas O2 flow rate,  is the applied RF power, and  is the O2 pressure in the chamber.”

Point 2: Errror. N + O → NO + O in the text...  - >  N + O_2

Response 2: Thank you for the comment. The erroneous chemical formula has been corrected in the revised manuscript.

Point 3: What is the value of the electron temperature measured using EEPF ? There is no information about it. Present the same figure as in case of the electron density. How does it impact the observed behaviour of the ashing rate.

Response 3: Thank you for the comment. We obtained the electron temperautre from the slopes of the EEPFs, which has been shown in Fig. 5 in the revised manuscript. The electron temperature of electrons with an energy higher than 7.5 eV, the threshold of oxygen excitation, is nearly constant at 2.4 eV. Since it barely changes with the N2 mixing ratio or RF powers, the electron temperature is thus expected to have little effect on the ashing rate.

Point 4: The OES measurements are line-of-sight integrated. What happens if the OES measurements would be performed not at the normal to wafer but along it surfaces ?

Response 4: Thank you for the comment. Along wafer surfaces, it is expected that the OES measurements would acquire much less emission signals, which might be insufficient to obtain oxygen radical information, since there is a lack of electrons near wafer surfaces that excites radicals. This is why we aligned the OES optical fiber to see the plasma bulk. We added a description on the OES alignment in the revised manuscript as below (as shown in blue in the revised manuscript).

“The optical fiber where line-of-sight integrated plasma emissions are detected is laterally mounted on the chamber wall and points the plasma bulk for strong signal measurement.”

Point 5: The ashing rate is increasing as a function of power. But at certain point the high voltage should have also a negative effect Explain it.

Response 5: Thank you for your professional comment. High voltages applied to the ICP antenna could have negative effects such as damage on the sample surface by increased ion energy. However, since increasing RF powers reduces capacitive coupling between the ICP antenna and the grounded chamber wall, the ion bombarding energy will in fact decrease as the RF power increases. We thus consider that under the range of RF powers employed in this work, such negative effects related to high RF powers are seemingly less.

Point 6: The results were obtained using the data at 8446 A.. What happens if another spectral line is used ? Have the authors performed it or not ?

Response 6: Thank you for the comment. In addition to the data at 8446 A, there is 7774 A, which is also well known for O radical emission. We have investigated the OES data using the 7774 A line, the comparison of which to the results obtained with the data at 8446 A is shown below.

Point 7: Whereas the topic is slightly different the similar approach is also used to combine the spectroscopic studies with ashing/sputtering rate in other RF plasmas... see for instance

Sackers et al, Physics of Plasmas 29, 043511 (2022); https://doi.org/10.1063/5.0083613

It would be important to demonstrate the connection between the different topics of plasma and surface physics.

Response 7: Thank you for the helpful comment. Following the comment, we added a discussion on the connection between the plasma parameters and surface chemistry in Section 3 as below (as shown in blue in the revised manuscript).

“It is worth noting that the PR used in this work contains C atoms in its components and thus oxygen radicals generated in plasmas are expected to an essential role in removing the PR coating layer from the wafer surface.”

Reviewer 2 Report

1. Please add more citations of OES and actinometry on O2 plasma. Some latest references are below:

Karakas, E., Donnelly, V. M., & Economou, D. J. (2013). Optical emission spectroscopy and Langmuir probe diagnostics of CH3F/O2 inductively coupled plasmas. Journal of Applied Physics113(21), 213301.

Li, H., Zhou, Y., & Donnelly, V. M. (2020). Optical and mass spectrometric measurements of dissociation in low frequency, high density, remote source O2/Ar and NF3/Ar plasmas. Journal of Vacuum Science & Technology A: Vacuum, Surfaces, and Films38(2), 023011.

2. Please describe more details about your experiment design and set-up. What's the total flowrate of O2 and N2? What's the range of mfc?

3. The reaction on line 115 should be "N+O+O ->NO+O". 

4. Could authors provide the explanation why the ashing rate peak point tended to move toward the higher ratio as the power increased? Will it keep moving toward the higher ratio with higher power?

5. Please provide zoom-in images from 3eV to 7eV on Figure 5.

6.From Figure 7, it seems that O density at 500W is lower than its at lower power. Please double check the number.

Author Response

Point 1: 1. Please add more citations of OES and actinometry on O2 plasma. Some latest references are below:

Karakas, E., Donnelly, V. M., & Economou, D. J. (2013). Optical emission spectroscopy and Langmuir probe diagnostics of CH3F/O2 inductively coupled plasmas. Journal of Applied Physics113(21), 213301.

Li, H., Zhou, Y., & Donnelly, V. M. (2020). Optical and mass spectrometric measurements of dissociation in low frequency, high density, remote source O2/Ar and NF3/Ar plasmas. Journal of Vacuum Science & Technology A: Vacuum, Surfaces, and Films38(2), 023011

Response 1: Thank you for the comment. Following the comment, we added the suggested references in the revised manuscript.

[33] Karakas, E., Donnelly, V. M., & Economou, D. J. (2013). Optical emission spectroscopy and Langmuir probe diagnostics of CH3F/O2 inductively coupled plasmas. Journal of Applied Physics113(21), 213301.

[34] Li, H., Zhou, Y., & Donnelly, V. M. (2020). Optical and mass spectrometric measurements of dissociation in low frequency, high density, remote source O2/Ar and NF3/Ar plasmas. Journal of Vacuum Science & Technology A: Vacuum, Surfaces, and Films38(2), 023011

Point 2: Please describe more details about your experiment design and set-up. What's the total flowrate of O2 and N2? What's the range of mfc?

Response 2: Thank you for the comment. The total flow rate of O2 and N2 is 150 sccm, and the range of MFC is 500 sccm and 200 sccm, respectively. Following the comment, we added this detailed description on our experimental setup in the revised manuscript (as shown in blue in the revised manuscript).

Point 3: The reaction on line 115 should be "N+O+O ->NO+O".

Response 3: Thank you for the comment. The erroneous chemical formula has been corrected in the revised manuscript.

Point 4: Could authors provide the explanation why the ashing rate peak point tended to move toward the higher ratio as the power increased? Will it keep moving toward the higher ratio with higher power?

Response 4: Thank you for your critical comment. The ashing rate peak occurs when the O2 density has also the maximum value [not applicable, however, to the 500 W case shown in Fig. 7(d). This may be attributed to other effects such as UV radiation from the plasma on the ashing rate]. With increasing RF powers, the ashing rate increases with the N2 mixing ratio due to an increase in the N2 molecules in the vibrational state and starts to decrease when the O2 molecule density becomes insufficient, as described in the manuscript. As the RF power increases, the electron population that can lead the N2 molecules to the vibrational state would increase further, which may result in the maximum ashing rate at higher N2 mixing ratio. The moving ashing rate peak however is considered to stop at some N2 mixing ratio point where the O2 portion reaches could no longer be less to achieve high ashing rates.

Point 5: Please provide zoom-in images from 3eV to 7eV on Figure 5.

Response 5: Thank you for the comment. Following the comment, we adjusted Fig. 5 as below in the revised manuscript (please note that the red lines have newly inserted in the Fig. 5 by following the reviewer 1 comment that requires the representation of the slopes of the graph where the electron temperature is obtained from).

Point 6: From Figure 7, it seems that O density at 500W is lower than its at lower power. Please double check the number.

Response 6: Thank you for the meticulous comment. We double checked the Fig. 7 and it seemed correct. The O radical density calculated from the OES results, shown in Fig. 7, is the value obtained with the density of high-energy electrons of over 7.5 eV that can excite O radicals. However, since the O radical density is in fact affect by the electron density that includes electrons with energy lower than the excitation threshold but higher than the dissociation threshold energy.